# Water Extraction of Tannins from Aleppo Pine Bark and Sumac Root for the Production of Green Wood Adhesives

**DOI:** 10.3390/molecules25215041

**Published:** 2020-10-30

**Authors:** Issam Dababi, Olinda Gimello, Elimame Elaloui, Nicolas Brosse

**Affiliations:** 1Laboratoire d’Etudes et de Recherche sur le Matériau Bois, Faculté des Sciences et Technologies, Université de Lorraine, EA 4370, Boulevard des Aiguillettes, BP 70239, 54506 Vandœuvre-Lès-Nancy CEDEX, France; dababiissam@yahoo.fr; 2ICGM, ENSCM, 240 Avenue du Professeur Emile Jeanbrau, 34296 Montpellier CEDEX 5, France; olinda.gimello@enscm.fr; 3Laboratoire Matériaux, Energie et Environnement, Faculté des Sciences de Gafsa, Université de Gafsa, 2100 Gafsa, Tunisia; limamealoui@gmail.com

**Keywords:** wood adhesive, tannins, hardeners, glyoxal, particleboard

## Abstract

The extraction of condensed tannins from Aleppo pine bark and sumac roots (*Brown Rhus tripartitum*) was examined in near industrial conditions, using a water medium in the presence of 2% NaHCO_3_ and 0.5% NaHSO_3_ at two different temperatures (70 °C and at 100 °C). The tannins extracts were recovered in high yields (~25% of Aleppo pine and ~30% for sumac) with high phenolic contents (>75%). The tannins were characterized by ^13^C-NMR and MALDI TOF and showed characteristics of procyanidin/prodelphinidin units. The tannins extracted at 100 °C were composed of smaller flavonoid oligomers (DP < 8) compared to those extracted at a lower temperature (DP > 10). Adhesive resin formulations were prepared using Aleppo or sumac tannins and four different cross linkers (hexamine, glutaraldehyde, furfural, and glyoxal). The resins were studied by TMA in bending and tannins-based formaldehyde-free wood particleboards were produced. The panels displayed internal bond strengths > 0.35 MPa with the four hardeners and all of them passed relevant international standard specifications for interior grade panels. The best results were observed with the tannins extracted at 70 °C with furfural as hardener (IB = 0.81 MPa for Aleppo pine and IB = 0.76 MPa for sumac).

## 1. Introduction

The production of environmentally friendly wood adhesives constitutes an area of active research since many years [1,2,3]. Tannin-based adhesives have been successfully described and used as particleboard panel binders [2]. Tannins can be classified into two major families, condensed and hydrolysable tannins [4]. Condensed tannins are primarily used for adhesive applications consist of flavonoid units with varying degree of polymerization. Figure 1 gives the general formula of flavonoid units with resorcinol (1 OH group in A ring) and phloroglucinol (2 OH groups in A ring) and catechol (2 OH, B ring) and pyrogallol (3 OH, B ring). Several cross-linkers have been experimented for the hardening of the tannin extracts including formaldehyde and hexamine [5,6,7]. Glyoxal, glutaraldehyde and *t*-phthaldialdehyde [8] have more recently been tested to substitute formaldehyde in order to limit formaldehyde emission and tannin-based formaldehyde-free adhesives have been used for interior and exterior wood bonding. However, the large-scale production of tannins-based adhesives is limited by the relativity limited supply of tannins.

Aleppo pine so called *Pinus halepensis Mill.* is a widespread tree of North Africa [9] and has been used traditionally for tannin production in the Middle East, Europe, and North Africa. In Tunisia, the Aleppo pine trees have been used mainly in wood production, resin tapping and seeds [10]. Sumac (*Rhus tripartitum*) is a dioecious shrub widespread in North Africa [11], especially in the steppes of desert, arid and semiarid areas. The most important part of this plant is the root bark due to its high content in tannins. The powdered root bark has been used to treat gastric ulcer, diarrhea, for dyeing leather and dye wool and silk [12,13,14].

Condensed tannins were extracted from Aleppo pine bark and sumac roots using methanol-based procedures followed by a liquid-liquid extraction with diethylether for characterization purposes [14]. It has been demonstrated that both substrates are especially rich procyanidin and prodelphinidin tannins and that they constitute potential for the production of wood adhesives [14,15]. However, the industrial development of such green wood adhesives depends on the availability of low-price, high-quality and high reactivity biopolymers. Industrial tannins are extracted in the form of non-purified mixtures by countercurrent extraction processes using hot water as solvent (between 70 °C and 90 °C) without organic solvents, but usually in presence of base (carbonate) and sodium sulfite or metabisulfite. The extraction yields using these conditions were around 15% of crude extract from pine bark and 30% for mimosa bark [16]. It is known that the extraction method considerably impacts the purity and the reactivity of condensed tannins [17]. Compared to solvent extracted tannins, the commercial tannins contain less active phenolic ingredients (50–80%), the non-tannins content consisting of water-soluble sugars and hydrocolloid gums. The non-tannin content strongly affects the strength and water resistance of the resulting adhesive. The utilization of sulfite facilitates the extraction of the high molecular weight oligomers by increasing their water solubility [18]. The temperature of extraction is also an important parameter. It has been previously observed that temperature around 100 °C promotes the extraction of the non-tannin components [19].

In this paper, we describe the extraction of tannins from Aleppo pine barks and sumac roots using a water-based process. The main goal of this work is to develop a green and industrially credible approach for tannin recovery from Mediterranean co-products. The tannin extracts have been characterized by nuclear magnetic resonance (NMR) and assisted laser desorption ionization-time of flight (MALDI-TOF) in order to identify their composition and structure. Adhesive formulations and wood panels have been produced and characterized using four different hardeners and without formaldehyde (furfural, glutaraldehyde, hexamine, and glyoxal).

## 2. Results and Discussion

The influence of the temperature on the extraction yields, gelation time (T-gel) and Stiasny number are summarized in the Table 1. Gelation times are an indication of the reactivity of the sites of tannins molecules contained in the extract toward formaldehyde and Stiasny numbers are an indication of the purity of the tannins extracts.

As shown in the Table 1 and compared with previous investigations [1,2], relatively high yields of crude tannins (>25% *w*/*w*) were obtained for both feedstocks. In accordance with recent reports and compared to other described species, Aleppo pine and Sumac root barks are rich in extractable tannins. Interestingly, high Stiasny numbers (>75%) were observed for the four extracts suggesting high polyphenolic purities. For Aleppo pine, the extraction yields and Stiasny numbers determined in this study were comparable or higher compared to those published for a solvent based tannins extraction [15]. Gel time provides valuable indications on the reactivity of tannin molecules toward cross-linkers and are highly dependent on pH. It is generally observed that the reactivity of tannins with formaldehyde is maximum at low pH (<2) and high pH (>7) and is the slowest around pH 4–5. The gel time values for the four extracts are given in Table 1 accompanied by the pH of the formulation. Taking into account the pH values (around pH 5), relative short gel time were observed demonstrating a high reactivity for all the extracts. As a result, Aleppo pine and sumac tannins could constitute a potential source of industrial tannins.

The ^13^C-NMR spectra of the extracts were performed in D_2_O (Figure 2) and were interpreted according to previous studies [3,20]. The spectra showed typical signals of condensed tannins. C5, C7, and C8a carbons of procyanidins were detected at ~155 ppm. Peaks at 143–145 ppm assigned to C3′ and C4′ and peak at 129–132 ppm to C1′. The broad peaks between 110 ppm and 95 ppm are assigned to C8, C6, C6′, and C2′. It has been previously reported that the relative intensities of the free C6 and C8 sites on A-ring at 95–98 ppm are a reliable indication of the chemical reactivity and of the degree of polymerization of the tannins [1,2,3]. Interestingly, the two tannins extracted at 100 °C (b and d) exhibited high intensity of these signals indicating a probable lower degree of polymerization (DP) compared to those extracted at 70 °C. A high intensity of the signal at ~105 ppm in spectra a and b, assigned to C4–C6 and C4–C8 interflavonoid linkages, confirmed the higher DP of the tannins extracted at the lower temperature. Thus, the lower DP of fractions b and d can be associated to a higher chemical reactivity toward cross-linking reactions [1,2,3]. A relatively high carbohydrate content could also be deduced by the high intensity of the sharp peaks at 60–85 ppm and around 100 ppm, the range characteristic of carbohydrates in tannin extracts.

Figure 3 shows the MALDI TOF analysis of tannin crude extracts of the two biomasses at 70 °C and 100 °C. In agreement with the ^13^C-NMR results, the MALDI-TOF MS analysis of the crude extracts from Aleppo pine and sumac between 160 *m*/*z* and 3000 *m*/*z* presents predominantly condensed tannins. The major mass increments were 272 *m*/*z* and 288 *m*/*z* and only few 306 *m*/*z* were detected (Figure 3). As proposed by Abdalla et al. [21] in condensed tannins the flavonoid units involved in the formation the oligomers were of three types indicated as A, B, and C corresponding to the masses 274 *m*/*z*, 290 *m*/*z* and 306 *m*/*z*, respectively. The molecular mass of 290 *m*/*z* can refer to two types of unit: robinetinidin and catechin, the structures of these types of flavonoids are presented in Figure 1. The combination of masses (A = 274, B = 290, C = 306) were used to calculated the masses of oligomers according to the following formula: *m*/*z* = 272 (A) + 288 (B) + 304 (C) + 23(Na) +2 (end groups, 2XH)
*m*/*z* 272, 288 and 304 correspond to the molecular weight of an extender fisetinidin unit of A, catechin (or epicatechin) unit of B and gallocatechin unit of C respectively.

In accordance with the NMR results previously discussed, the temperature of the tannin extraction impairs the DP of oligomers. The tannins extracted at 70 °C displayed the highest molecular mass, while homopolymers with type A units in the form of sodium adducts up to tetramers and type B tannin units up to hexamers were observed. Heteropolymers resulting of the combination of AB units up to decamers were detected with a maximum obtained from sumac at 70 °C (2796 *m*/*z*). Regarding the extracts at 100 °C, heteropolymers with lower DP, ranging from 2 to 7 flavanoid units, were detected.

The values obtained by the thermo-mechanical analysis enable a good prediction of the behavior of an adhesive mix for particleboard production. 16 adhesive resin formulations were prepared using the Aleppo or sumac tannin solutions (extracted at 70 °C and 100 °C) and four different cross linkers: hexamine, glutaraldehyde, furfural and glyoxal. The formulations were scanned by thermomechanical analysis in bending as previously reported [3,14,22]. The evolution of moduli of elasticity (MOE) with the temperature are given in Figure 4A,B.

The MOE max observed for all the formulations, measured between 4000 MPa and 7000 MPa, were high compared to the data from literature [23]. This result suggests that Aleppo and sumac tannins extracted in a water medium gave high wood-joint strengths whatever the hardener used.

The reactivity of the four cross-linkers were in the following order: furfural > glyoxal > hexamine > glutaraldehyde. It also appeared from the TMA study that the temperature of the tannins extraction (70 °C or 100 °C) and therefore the tannins DP had generally a slight or no effect on the resins performance, except for glyoxal as hardener which gave significantly higher MOE for Aleppo tannins extracted at 100 °C and for sumac tannins extracted at 70 °C. For Aleppo pine resins (Figure 4B), an additional signal is observed at ~100 °C when glutaraldehyde is used as cross-linker. This signal can be assigned to a first cross-linking reaction through formation of a non-stable cross linkage. This behavior has been previously reported [24].

The different tannin-based adhesive formulations studied were used to press one-layer wood particleboards. The internal bond (IB) of the panels which are a direct measure of the performance of the adhesives were established. The results are shown in the Figure 5. It appears that the panels produced displayed good internal bond strengths with the four hardeners and passed relevant international standard specifications for interior-grade panels (IB > 0.35 MPa). The results obtained were generally in accordance with a previous TMA experiment. As shown by TMA, the utilization of glutaraldehyde as hardener yielded the low IB (~0.4 Mpa). This result is in accordance with a previously published paper which assessed the impact of aldehyde chain length on the tannins-based resin performance [25].

## 3. Materials and Methods

### 3.1. Materials

Aleppo pine bark and *brown Rhus tripartitum* (sumac) bark were harvested in the area of Gafsa (Tunisia). The barks were air-dried at room temperature and then cut into small particles (1 mm). Before analysis, the samples were stored at 5 °C until needed for analysis. All chemicals used were of analytical or reagent grade (Sigma-Aldrich, Saint Louis, IN, USA).

### 3.2. Extraction of Tannins

The barks (100 g) were treated with one liter of an aqueous solution of base (NaHCO_3_ 0.5%, NaHSO_3_ 2%). Samples were immersed in water under continuous magnetic stirring for 6 h at 70 °C or 100 °C according to published procedures [17,18,19,20,21,22,23,24,25,26]. The raw tannin extracts were filtered using a fritted glass and dried in an oven at 50 °C. The extraction rate was calculated according to following Equation (1): (1)$$\mathrm{Extraction}\;\mathrm{rate}\;(\%)\;=\;\frac{Dried\;weight\;of\;precipate}{Sample\;dry\;weight\;}\times \;100$$

### 3.3. Determination of Stiansy Number

According to Yazaki [17], the Stiasny number was used to determine the reactivity of tannin toward formaldehyde. Briefly, 0.2 g (oven-dry-mass) of tannin, 5 mL of 37% formaldehyde and 5 mL of 10 M hydrochloric acid solution were mixed and heated under reflux for 30 min. The tannin extracts were recovered after precipitation and dried to a constant weight in an oven at 105 °C.

The Stiasny number was calculated and determined as follows Equation (2):(2)$$\mathrm{Stiasny}\;\mathrm{number}\;(\%)\;=\;\frac{Dried\;of\;precipated\;sample}{Dissolved\;sample}\times \;100$$

### 3.4. Gelation Time (T-gel)

The gelation time is defined as the time required for tannin/formaldehyde from a colloidal solution to become a solid or semi-solid jelly or gel. According to previous works [20], an aqueous solution of 45% of tannin was placed in a glass test tube. Then, 5% of paraformaldehyde (based the weight of tannin) was added and held in a water bath at temperature of 100 °C. The time taken to reach the gel point was recorded during constant stirring with the aid of a wire spring and a stopwatch. The test was duplicated three times. The pH of the mixture was measured using a microprocessor pH meter (Hanna pH 211).

### 3.5. Characterization of Tannin Extracts

#### 3.5.1. NMR Analysis

NMR analysis was carried out on the ^13^C atom, in liquid phase (four samples). The spectra were recorded on a Brükeravance 400 MHz spectrometer (Billerica, MA, USA). The chemical shifts were calculated relative to TMS (tetramethylsilane). The spectra were recorded at 100.6 MHz. The number of scans and the acquisition time were 12,000 and 1.36 s respectively. Tannin extract (70 mg) was dissolved in 0.5 mL D_2_O (deuterated water). Samples were placed in 5 mm NMR tubes (sample volume of 0.5 mL).

#### 3.5.2. MALDI-TOF Mass Analysis

MALDI-TOF-MS spectra were acquired with a MALDI-TOF/TOF BrukerUltraflex III mass spectrometer (Billerica, MA, USA) using a nitrogen laser for MALDI (λ = 337 nm). Mass spectra of 2500 shots were accumulated for the spectra at 25 kV acceleration voltage and reflectron lens potentials at 26.3 KV. Peptides have been used for calibration. The four tannins extracts were dissolved at 20 mg/mL in water:acetone (50:50, *v*/*v*). DHB (2,5-dihydroxybenzoic acid) was used as matrix. It was dissolved in acetone:water (50:50 vol%, 10 mg/mL). A mixture composed of 4 µL of tannin sample, 10 µL of matrix solution and 1 µL of salt were hand spotted on a MALDI target.

#### 3.5.3. Resin Adhesive Preparation

Aleppo pine and sumac tannin water solutions (45%, *w*/*w*) were prepared by dissolving the dried powder of each tannin extract in water. The pH was adjusted to 10 by addition of NaOH 33% water solution. A water solution (45%, *w*/*w*) of hexamine (6%, *w*/*w* on a solid tannin basis), furfural (7%, *w*/*w* on a solid tannin basis), glutaraldehyde (11%, *w*/*w* on a solid tannin basis), or glyoxal (9%, *w*/*w* on a solid tannin basis) were added to the mixture while stirring [1,8].

#### 3.5.4. Thermomechanical Tests

Triplicate samples of two beech wood ply each 0.6 mm thick, bonded with each formulation, for a total sample dimension of 21 × 5 × 1.2 mm were tested under atmospheric atmosphere between 25 °C and 250 °C at a heating rate of 10 °C/min and for 30 mg of tannin-based resin using a Mettler Toledo TMA40 in three-point bending. A force varying continuously between 0.1 N, 0.5 N, and back to 0.1 N was applied on the specimens with each force cycle of 12 s (6 s/6 s).

All tests were performed by TMA under the following conditions. Heat rate = 10 °C/min, 22 mg of tannins-based resin, in the temperature range of 25–250 °C. The thermomechanical analyzer used was a Mettler Toledo TMA40 [8].

#### 3.5.5. Particleboard Manufacture and Testing

Particleboards of 350 × 300 × 16 mm^3^ (one-layer) were prepared in triplicate using tannins-based resins previously described and wood particles (*Fagus sylvatica* and *Picea abies*) at 3.43 MPa with maximum pressure and 190–195 °C press temperature according to a previously described method [20]. The adhesive resin was 10% based on dry wood. The total pressing time was 7.5 min. The particleboards were tested for dry internal bond (IB) strength test, which is a relevant international standard test (EN 312).

## 4. Conclusions

Tannin-rich crude extracts were isolated from Aleppo pine and sumac barks using experimental conditions similar to those used industrially (aqueous medium at 70–100 °C). It has been shown by NMR and MALDI-TOF that the tannins contained in the isolated extracts showed characteristics of procyanidin/prodelphinidin units. Adhesive resin formulations and wood panels were prepared using four different cross linkers (hexamine, glutaraldehyde, furfural, and glyoxal). All the panels produced displayed very good internal bond strengths (>0.35 MPa) with the four hardeners and passed relevant international standard specifications for interior grade panels. We hope that this work can contribute to a better use of low-valued Mediterranean plant co-products.

## Figures and Tables

**Figure 1 molecules-25-05041-f001:**
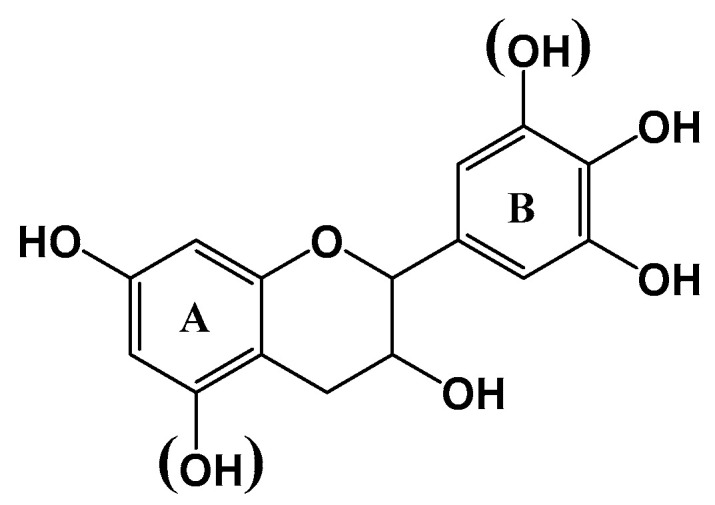
Flavonoid units involved in the formation of oligomers.

**Figure 2 molecules-25-05041-f002:**
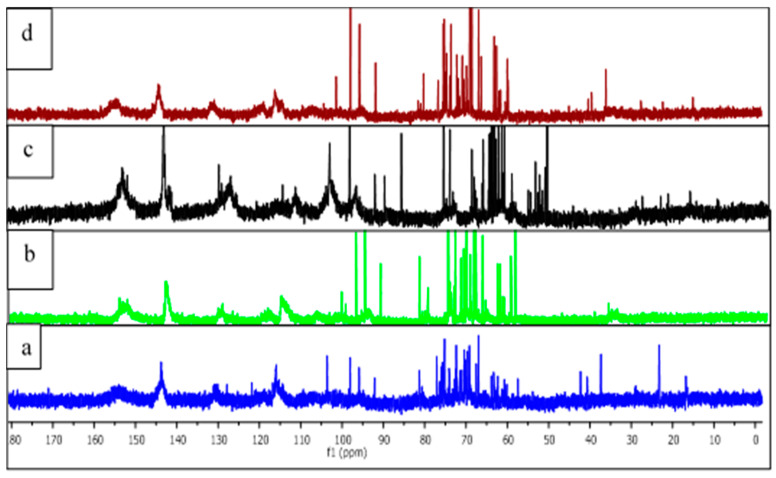
^13^C-NMR spectra of different tannins from Tunisian plants. (**a**) Aleppo pine 70; (**b**) Sumac 70; (**c**) Aleppo pine 100; (**d**) Sumac 100.

**Figure 3 molecules-25-05041-f003:**
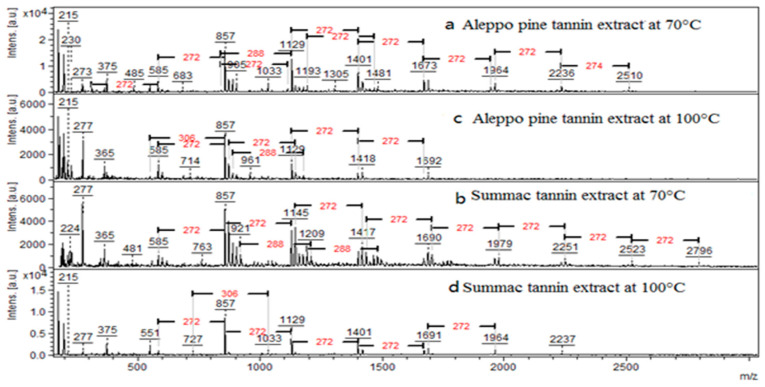
MALDI-TOF mass spectra of Aleppo pine extracts (**a**,**c**) and sumac extracts (**b**,**d**) obtained in positive ion mode between 160 *m*/*z* and 3000 *m*/*z*.

**Figure 4 molecules-25-05041-f004:**
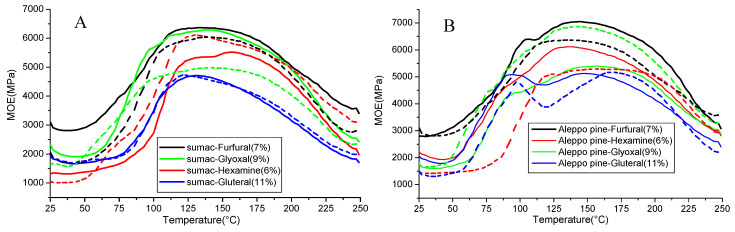
Comparison of thermo-mechanical analysis curing curves of tannins-based resins; continuous line: tannin extract at 70 °C; dashed line: tannin extract at 100 °C. (**A**) Sumac tannins, (**B**) Aleppo tannins.

**Figure 5 molecules-25-05041-f005:**
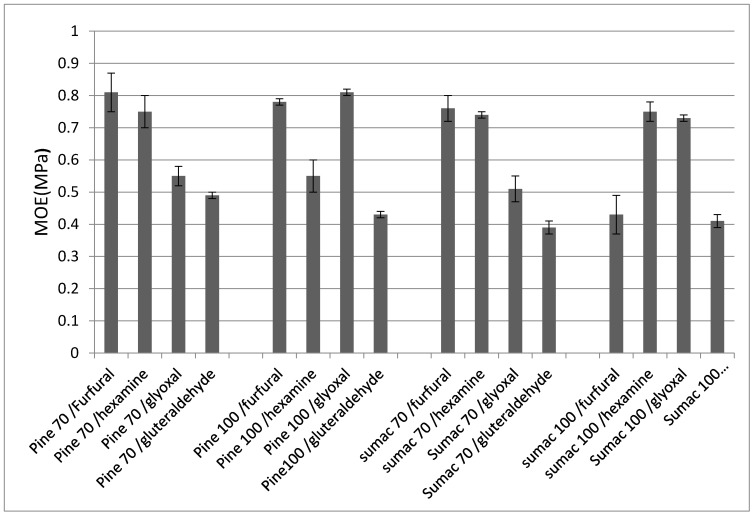
Internal bond strength (IB) of tannins-based particleboards. 70 and 100 refer to the extraction tempature of the tannins.

**Table 1 molecules-25-05041-t001:** The yield, gelation time (Tg), Stiasny number, total polyphenols of the different extracts from plant.

Essay	Species	Condition ^a^	Yield (%) ^b^	Stiasny Number % ^b^	Gel. Time (s)	pH_0_
1	Aleppo pine	T = 70 °C	25 ± 1.2	83 ± 0.2	100	4.88
2	Aleppo pine	T = 100 °C	26 ± 0.9	81 ± 1.4	105	5.1
3	Sumac	T = 70 °C	29 ± 2.2	75 ± 2.2	130	5.23
4	Sumac	T = 100 °C	32 ± 1.2	78 ± 0.9	119	5.44

^a^ water, 2% NaHCO_3_; 0.5% NaHSO_3_; ^b^ average of 3 independent extractions (with standard deviation).

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
