# Peer review of "Water Extraction of Tannins from Aleppo Pine Bark and Sumac Root for the Production of Green Wood Adhesives"

_molecules, 2020, doi:10.3390/molecules25215041_

Round 1

Reviewer 1 Report

Please see uploaded file below

Author Response

Thank you for the detailed correction and constructive remaks

The corrections have been done and are highlighted in yellow in the text. 

The conclusion has been re writted

Reviewer 2 Report

The title anf the experimentals done do not correlate. In additon, based on the results the conclusions made are not valid.

Please, look the comments as a separate attachment.

Author Response

We improved the manuscript on the basis on this report and most of the modifications have been done accordingly. Modifications are highlighted in blue.

The main corrections are :

- a better explanation in the text regarding the isolation of a raw mixture containing tannins.

- The fact that the tannins compounds have not been separated but characterized and used in mixtures

- Explanation + 1 additional reference about Fig. 4 (peak at 100°C)

- addition fig 5 about panels production

- the conclusion was re-writted

Reviewer 3 Report

The work is potentially interesting; however, I have several concerns about the work carried out.

In general, the language throughout all the manuscript should be strongly improved, as well as the English grammar that is wrong in many sentences.

In the material and methods a lot of important information is missing

In addition, there is not any statistical treatment of the data.

Please see the enclosed file for detailed comments.

Author Response

The corrections have been done and are highlighted in green in the text. 

Reviewer 4 Report

Have some questions, corrections, suggestions, etc. Read enclosed file, please.

Author Response

The corrections have been done and are highlighted in purple in the text

Round 2

Reviewer 2 Report

Title: Water extraction of Aleppo pine bark and sumac root tannins for the production of green wood adhesivesà should be

Water extraction of tannins from Aleppo pine bark and sumac root for the production of green wood adhesives

Other comments:

Abstract

Lines 15-16: The text “… in the presence of 2% NaHCO3, 0.5% NaHSO3 at 70°C and at 100°C.” à The sentence is not informative. The authors need to explain when the alkaline solution and the acidic solution was used. When were the temperatures selected? They cannot be used simultaneously.

Line 24: The pressure should be “> 0.35 MPa”

Extraction of tannins

Chapters 2.1, 2.2, 2.3, and 2.4 and 2.5.5 repeat the existing procedures and methods. It seems that there has not any really new developed. What do the authors think about the novelty value is in the experimental part of the work?

How could 100 g of bark be extracted with magnetic stirring? à The authors need to explain the instrumentation.

Line 86: What does the sentence mean?  “….solid-to-liquid ratio of 1:10 (w/v)? à Does it mean 100 g wood + 1L solvent? OR what?

Line 93: What is the solvent in 50 mL of tannin (0.4 %, w/w) solution? & What is the 0.4 % (w/w) sample? What components were weighted?

Lines 102-103: For how long time were “5% of paraformaldehyde (based the weight of tannin) and tannin reacted in a water bath at temperature of 100°C.”?

Lines 131-133: About the TMA instrument: What was the inert gas used? What is the name of the 22 mg resin?

Results

Table 1: How was pH0 measured?

Table text should be above the table, not below.

Line 164: What do the authors mean? “The solution 13C NMR spectra of the four extracts…” à What solutions were studied with 13C NMR?

Lines 170-172: Extraction made at 70°C & 100oC: Were the compounds extracted at both temperatures? à The authors make a conclusion that the analytes were not polymerized at 70°C, but were polymerized at 100°C? -à Have the authors considered that at the higher temperature the analytes may also be degraded, not perhaps at all polymerized?

The gel formation is not informed as it should. The authors should explain it deeply, since it is an essential material product in the study presented.

The sub-figures in Figure 2 are very noisy (background). In addition, all signals in the figures need explanation.

Lines 199-203 need to be sentences, not a list. If the sentences are not possible to write, the information needs to be in a table. Thus, the authors can list all the MALDI-TOF MS data in one table.

Line 212: Figure 3: Can the authors identify what tannins are in the spectral data? In addition, the caption of the text needs to inform the amount of the extracts used to creating the figures. How much was introduced to MALDI-TOF?

Line 218: The text is “thermomechanical analysis in bending as previously reported.” à A reference is needed.

Figure 4: What is the difference of the samples in A and B?  What were the weights of the modified samples in the tests? What are the main components in A. Sumac tannins, B. Aleppo tannins extracts?

  • mass spectrometric identification is needed.

Line 243: How was the “glutaraldehyde containing hardener yield with the low IB (~0.4 MPa)” measured and calculated?

The authors have not measured the matrix effect in the samples. In addition, they have not considered the case that there are a lot of sugars and organic acids and VOC in the extracts, as well.

Conclusions

Usually the industry partners are interested in pure extracts, not mixtures.  How do the authors think that the mixture of tannins and matrix compounds would be interesting enough for material production?

Author Response

Reviewer 2

We improved the manuscript on the basis on this report and most of the modifications have been done accordingly. Modifications are highlighted in blue.

The main corrections are :

- a better explanation in the text regarding the isolation of a raw mixture containing tannins.

- The fact that the tannins compounds have not been separated but characterized and used in mixtures

- Explanation + 1 additional reference about Fig. 4 (peak at 100°C)

- addition fig 5 about panels production

- the conclusion was re-writted

Detailed responses :

Abstract

Lines 15-16: The text “… in the presence of 2% NaHCO3, 0.5% NaHSO3 at 70°C and at 100°C.” à The sentence is not informative. The authors need to explain when the alkaline solution and the acidic solution was used. When were the temperatures selected? They cannot be used simultaneously.

Response :  The sentence has been modified. Sulfite and hydrogenocarbonate ions are indeed used simultaneously

Line 24: The pressure should be “> 0.35 MPa”

Response :  modified accordingly

How could 100 g of bark be extracted with magnetic stirring? à The authors need to explain the instrumentation.

Response :  We confirm that a magnetic stirrer was used

Line 86: What does the sentence mean?  “….solid-to-liquid ratio of 1:10 (w/v)? à Does it mean 100 g wood + 1L solvent? OR what?

Response :  the sentence was modified

Line 93: What is the solvent in 50 mL of tannin (0.4 %, w/w) solution? & What is the 0.4 % (w/w) sample? What components were weighted?

 Response : the sentence has been modified

Lines 102-103: For how long time were “5% of paraformaldehyde (based the weight of tannin) and tannin reacted in a water bath at temperature of 100°C.”?

 Response : the sentence has been modified

Lines 131-133: About the TMA instrument: What was the inert gas used? What is the name of the 22 mg resin?

 Response : “under atmospheric atmosphere”  was added. “Resin” was substituted by “tannin-based resin” for a better understanding

Results

Table 1: How was pH0 measured?

Response : “The pH of the mixture was measured using a microprocessor pH meter (Hanna pH 211)” was added in section 2.4

Table text should be above the table, not below.

 Response : modification done

Line 164: What do the authors mean? “The solution 13C NMR spectra of the four extracts…” à What solutions were studied with 13C NMR?

Response :  the sentence was modified for clarification

Lines 170-172: Extraction made at 70°C & 100oC: Were the compounds extracted at both temperatures? à The authors make a conclusion that the analytes were not polymerized at 70°C, but were polymerized at 100°C? -à Have the authors considered that at the higher temperature the analytes may also be degraded, not perhaps at all polymerized?

 Response : In this section, we only discussed the DP (molecular weight) of the extracted tannins

Lines 199-203 need to be sentences, not a list. If the sentences are not possible to write, the information needs to be in a table. Thus, the authors can list all the MALDI-TOF MS data in one table.

  Response : modification done

Line 212: Figure 3: Can the authors identify what tannins are in the spectral data? In addition, the caption of the text needs to inform the amount of the extracts used to creating the figures. How much was introduced to MALDI-TOF?

Response : The identification of the spectra are given in the text.  The amount of tannin is given in M&M section (2.5.2)

Line 218: The text is “thermomechanical analysis in bending as previously reported.” à A reference is needed.

 Response : references were added

Figure 4: What is the difference of the samples in A and B?  What were the weights of the modified samples in the tests? What are the main components in A. Sumac tannins, B. Aleppo tannins extracts?

Response : Signifiaction of A and B is in the fig caption. The quantities and conditions of the test are given in M&M section 

The authors have not measured the matrix effect in the samples. In addition, they have not considered the case that there are a lot of sugars and organic acids and VOC in the extracts, as well.$

Response : The production of this kind of green adhesive is based on the utilization of crude extracts. The purity (sugar content) was evaluated (15-25%) with the stiasny numbers given in Table 1

Conclusions

Usually the industry partners are interested in pure extracts, not mixtures.  How do the authors think that the mixture of tannins and matrix compounds would be interesting enough for material production?

Response : Such tannins based adhesives are currently commercialized in South Africa (industrial production from Quebracho tannins).

Reviewer 3 Report

some other changes are still required.

Line 80: How many samples did you collect?

Line 119 and 125: put the references at the end of the paragraph (not at the the end of the titles).

Line 136: put Fagus sylvatica and Picea abies in italic

Line 144-146. Again, this is materials and methods

Table 1: Again, there is not any statistical treatment of the data (just the standard deviations are not enough). This is very important for the significance and for the comparison of your results.

Line 149: replace "of the different extracts from plant" with " of the different plant extracts "

The quality of the figures is very low (in Fig. 2 and 3 is not possible to read the numbers; in Fig. 4 the text f is too small and in Fig. 5 the y axes is not readable and the outlines of the figures are missing).

Author Response

The corrections have been done and are highlighted in green in the text. 

Figures have been improved.

Regarding the data treatment, we gave the mean of 3 independent extraction with standard deviations. This type of presentation is very generally accepted. 

Reviewer 4 Report

Have a some questions, corrections, suggestions, etc. Read enclosed file, please.

Author Response

corrections have been done
